# An Empirical Study on the Efficacy of Deep Active Learning Techniques

## Abstract

Deep Active Learning (DAL) has been advocated as a promising method to reduce labeling costs in supervised learning. However, existing evaluations of DAL methods are based on different settings, and their results are controversial. To tackle this issue, this paper comprehensively evaluates 19 existing DAL methods in a uniform setting, including traditional fully-supervised active learning (SAL) strategies and emerging semi-supervised active learning (SSAL) techniques. We have several non-trivial findings. First, most SAL methods cannot achieve higher accuracy than random selection. Second, semi-supervised training brings significant performance improvement compared to pure SAL methods. Third, performing data selection in the SSAL setting can achieve a significant and consistent performance improvement, especially with abundant unlabeled data. Our findings produce the following guidance for practitioners: one should (i) apply SSAL as early as possible and (ii) collect more unlabeled data whenever possible, for better model performance. We will release our code upon acceptance.

## 1 Introduction

Training a well-performed Deep Neural Network (DNN) generally requires a substantial amount of labeled data. However, data collection and labeling can be quite costly, especially for those tasks that require expert knowledge (e.g., medical image analysis (Hoi et al.) and malware detection (Nissim et al., 2014)). Deep Active Learning (DAL) has thus long been advocated to mitigate this issue, wherein we proactively select and label the most informative training samples. That is, given a pool of unlabeled data, DAL iteratively performs data selection and training until the given labeling budget is reached, as shown in Figure 1.

Various DAL techniques are proposed in the literature. Most of them are fully-supervised ones (SAL) and aim for a better data selection strategy[1]. The SAL strategies can be roughly grouped into three categories: 1) *model-based selection*; 2) *data distribution-based selection*; and 3) *hybrid selection*. Model-based selection prefers annotating data that are most uncertain under the task model (Gal et al., 2017; Beluch et al., 2018). Data distribution-based methods select data according to their density or diversity (Sener & Savarese, 2018; Sinha et al., 2019). Hybrid methods consider task model information and data distribution when selecting (Ash et al., 2020).

By applying pseudo-labels (Arazo et al., 2019) to the unlabeled data or consistency regularization (Berthelot et al., 2019), semi-supervised learning (SSL) can improve the model performance substantially. Consequently, it is attractive to apply active learning on top of SSL techniques, referred to as semi-supervised active learning (SSAL). Song et al. (2019) incorporate the well-known SSL method MixMatch (Berthelot et al., 2019) during training. Gao et al. augment the unlabeled samples and enforce the model to have consistent predictions on the unlabeled samples and their corresponding augmentations. Also, Gao et al. develop a data selection strategy for the SSL-based method, i.e., selecting samples with inconsistent predictions. Besides, WAAL formulates the DAL as a distribution matching problem and trains the task model with an additional loss evaluated from the distributional difference between labeled and unlabeled data (Shui et al., 2020).

Despite the effectiveness of existing methods, the results from previous works often contradict each other. For example, CoreSet (Sener & Savarese, 2018) and the DBAL method (Gal et al., 2017) are

---

[1]Among the 19 investigated methods, 14 of them are fully-supervised ones

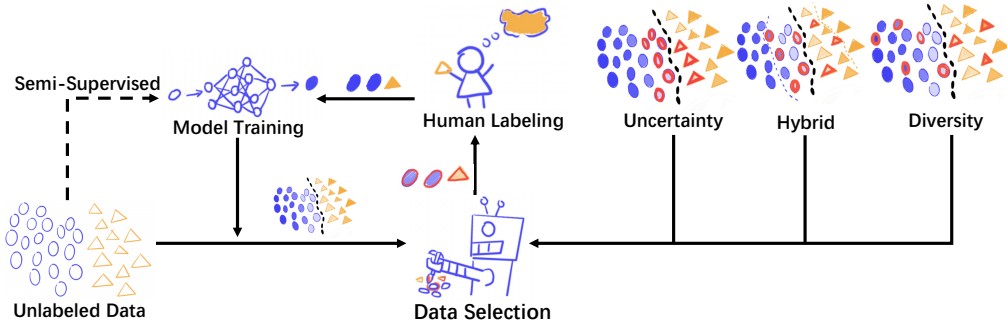

Figure 1: The pool-based deep active learning process. In each iteration, we first train the task model. Based on the trained model and the available unlabeled data, we select a subset of the unlabeled data for labeling (marked as red circle or triangle). The process is iterated until certain model accuracy is achieved or the labeling budget is used up. Existing methods mainly focus on the data selection strategy or the model training strategy.

shown to perform worse than the random selection (RS) in (Beluch et al., 2018) and (Sinha et al., 2019), respectively. Such result inconsistency is often due to the different experimental settings in evaluation (see Appendix Table 5). Several empirical studies are therefore proposed to address this problem (Beck et al., 2021; Munjal et al., 2020). However, again, they have controversial observations. Beck et al. (2021) claim that there is little or no benefits for different DAL methods over RS, while Munjal et al. (2020) show that DAL methods are much better than RS.

These inconsistencies motivate us to unify the experimental settings and conduct a thorough empirical study on the effectiveness of DAL techniques. Our contributions are summarized as follows:

- We re-implement and perform extensive evaluations of 19 deep active learning methods on several popular image classification tasks, including MNIST, CIFAR-10, and GTSRB. To the best of our knowledge, our evaluation is the most comprehensive one, which not only includes most state-of-the-art SAL solutions but also incorporates various SSAL methods.

- Through extensive experiments, we conclude that SSAL techniques are preferred. Traditional SAL methods can hardly beat random selection, and there is no SAL method that can consistently outperform others. In contrast, SSAL methods not only easily outperform all the SAL methods by a large margin. More importantly, active sample selection plays an important role in SSAL methods, which can achieve significant and consistent performance improvements compared to random selection.

- We conduct an in-depth analysis of SSAL methods and provide two guidance to the practitioners. First, one should conduct SSAL as early as possible. Second, one should seek more unlabeled data whenever possible to achieve better performance.

The rest of the paper is organized as follows. Section 2 presents the related works. Section 3 illustrates the experimental setup. We present our empirical study on the performance in Section 4. Further studies on SSAL methods are presented in Section 5. Section 6 concludes this paper.

## 2 RELATED WORKS

In this section, we introduce existing DAL methods and empirical studies on these DAL methods. As shown in Figure 1, existing DAL works can be roughly grouped into fully-supervised active learning (SAL) and semi-supervised active learning (SSAL), depending on whether they use unlabeled data to train the task model.

### 2.1 FULLY-SUPERVISED ACTIVE LEARNING (SAL)

We categorize the SAL strategies into three classes: uncertainty-based selection, diversity/ representativeness-based selection, and hybrid selection, which is the combination of the above two methods.

**Uncertainty-based selection.** The uncertainty-based approaches estimate the uncertainty for each unlabeled sample based on the task model's output and select uncertain data points for labeling. Various methods have been developed to estimate uncertainties. Least Confidence, Entropy Sampling, and Margin Sampling directly use the model output for an input image for sample uncertainty estimation (Wang & Shang, 2014). Specifically, Least Confidence calculates the uncertainty of a sample as $1 - largest\ output\ probability$, Entropy Sampling uses the output entropy to represent the uncertainty, and Margin Sampling uses the difference between the largest and the second largest output probability as the uncertainty of a sample. BALD Dropout runs the task model multiple times for an unlabeled sample with a certain dropout rate. Data uncertainty is defined by the difference between the mean entropy of multiple probability predictions and the entropy of the mean probability prediction (Gal et al.). Learning Loss trains an additional module to predict the loss of the original model and regards the loss as sample uncertainty (Yoo & Kweon, 2019). Uncertain GCN uses a graph convolution network (GCN) to select samples that are unlike the labeled samples (Caramalau et al., 2021). VAAL trains a variational auto-encoder (VAE) and a discriminator to distinguish labeled data and unlabeled data through adversarial learning (Sinha et al., 2019). They regard the probability of belonging to the unlabeled dataset as the uncertainty of a sample.

**Diversity/Representativeness-based selection.** Diversity/Representativeness-based methods aim at finding a diverse or representative data set from the unlabeled data pool such that a model learned over the subset is competitive over the whole dataset. CoreSet maps the high-dimensional data to low-dimensional feature space and based on which they select the $k$-center points (Sener & Savarese, 2018). KMeans clusters data points into multiple clusters based on their features and selects samples that are closest to the center of each cluster (Zhdanov, 2019). CoreGCN is a variant of uncertainGCN which applies the CoreSet algorithm on the feature learned by the GCN (Caramalau et al., 2021).

**Hybrid selection.** In practice, DAL methods typically acquire a batch of data at once. However, on the one hand, samples selected by the uncertainty-based method may contain redundant information. On the other hand, selecting samples without considering the task model's feedback can select samples that the model already accurately infers, thus providing little information to the model. To address this challenge, some methods propose to combine the uncertainty and diversity methods. ActiveLearningByLearning (ALBL) proposes to blend different AL strategies and select the best one by estimating their contribution at each iteration (Mayer & Timofte, 2020). BadgeSampling leverages the estimated gradient of the parameters in the last layer for selection. The selected samples have high gradient magnitude and diverse gradient directions (Ash et al., 2020). ClusterMargin clusters the unlabeled data and uses a round-robin strategy to select each cluster's most uncertain data (Citovsky et al., 2021). MCADL considers multiple criteria simultaneously and adaptively fuses the result of uncertainty-based selection and density and similarity-based selection (Yuan et al., 2019).

## 2.2 SEMI-SUPERVISED ACTIVE LEARNING (SSAL)

Another group of works that address the annotation-efficient learning problem is semi-supervised learning (SSL). The main idea of SSL is to mine the useful information in the unlabeled data for model training. One way is to assign each unlabeled sample a pseudo-label and then use these pseudo-labeled samples together with labeled samples for training. Another way is to use consistency regularization evaluated on the unlabeled data for better model training.

Recently, there have been few attempts to combine SAL and SSL to form the semi-supervised active learning methods (SSAL). Song et al. (2019) incorporates one of the popular SSL techniques MixMatch, into the SAL training pipeline, and the model accuracy is greatly improved. They use two sample selection algorithms, and we denote them as SSLDiff2AugDirect and SSLDiff2AugKmeans:

- **SSLDiff2AugDirect**. The first step of this method is to calculate the average prediction for each sample and its augmentation. Based on the averaged prediction, the authors use the difference between the largest and the second largest probability as the uncertainty for sample selection.

- **SSLDiff2AugKmeans**. They use the k-means algorithm to cluster all unlabeled samples. Then, they select the top uncertain samples from each cluster. The uncertainty is calculated as the SSLDiff2AugDirect method.

Table 1: Comparison of empirical studies of DAL methods.

| Paper | Publication Year | Methods | SSL | Performance | Robustness | Cost |
|---|---|---|---|---|---|---|
| Ours | 2022 | 19 | Yes | Yes | Yes | Yes |
| Beck et al. (2021) | 2021 | 8 | No | Yes | Yes | Yes |
| Hu et al. (2021) | 2021 | 12 | No | Yes | Yes | No |
| Munjal et al. (2020) | 2020 | 6 | No | Yes | Yes | No |
| Mittal et al. (2019) | 2019 | 6 | Yes | Yes | No | No |

Another SSAL method (we denote it as **SSLConsistency**) uses consistency regularization to improve the training process. Specifically, for each unlabeled sample, the authors enforce the model to produce consistent predictions on the sample and its augmented version. SSLConsistency also proposes a sample selection mechanism based on this consistency and selects the least consistent ones to label (Gao et al.). Please note that we also include the WAAL method as an SSAL because it uses unlabeled data to train the task model. Specifically, WAAL enhances the task model's feature extractor by letting it to differentiate the labeled and unlabeled empirical distributions based on the Wasserstein distance. It selects unlabeled samples that are different from the current labeled data.

## 2.3 EMPIRICAL STUDY OF ACTIVE LEARNING TECHNIQUES

Several empirical studies on the effectiveness of DAL methods have been proposed in recent years (Beck et al., 2021; Hu et al., 2021; Munjal et al., 2020; Mittal et al., 2019). We summarize their characteristics in Table 1. Recent empirical studies either fail to study SAL methods under a variety of experiment settings or fail to cover state-of-the-art methods. Specifically, most of them ignore a growing line of work that combines semi-supervised learning with active learning. While (Mittal et al., 2019) compares the performances of SSAL and SAL, it does not study the characteristics of SSAL methods further.

In contrast, our work unifies the SAL and SSAL into a single evaluation framework. Also, we provide an in-depth analysis of the SSAL methods. This analysis provides practical guidance to the users.

## 3 EXPERIMENT SETUP

**DAL techniques under study.** We re-implement and reproduce the results for 19 DAL methods, including the fully-supervised active learning (SAL) and the semi-supervised active learning (SSAL). For SAL methods, we cover **RandomSampling**, **ActiveLearningByLearning**, **AdversarialBIM**, **BALDDropout**, **BadgeSampling**, **ClusterMarginSampling**, **CoreSet**, **KMeansSampling**, **LearningLoss**, **LeastConfidence**, **MCADL**, **CoreGCN**, **UncertainGCN**, and **VAAL**. For SSAL methods, we cover **SSLRandom**, **SSLConsistency**, **SSLDiff2AugDirect**, **SSLDiff2AugKmeans**, and **WAAL**. For a detailed description of each method, please refer to Section 2.

**Datasets.** In this paper, we use three widely used image classification datasets for evaluation: MNIST (LeCun et al., 1998), CIFAR10 (Krizhevsky et al., 2009), and GTSRB (Houben et al., 2013). MNIST contains 70,000 examples of handwritten digits. The size of each image is $28 \times 28 \times 1$. CIFAR10 is composed of ten classes of natural images and has 60,000 images in total. The size of each image is $32 \times 32 \times 3$. GTSRB contains 43 classes of traffic sign images. There are 39,209 labeled images in total. Each image has $32 \times 32 \times 3$ pixels. Please note that the GTSRB dataset has imbalanced classes.

**Training settings.** We evaluate all methods with a unified setting, which is adopted from (Ash et al., 2020). Specifically, for MNIST, we set the start budget (denoted as nStart) as 2%, the total budget (denoted as nEnd) as 20%, the step size (denoted as nQuery) as 2%, and the training epochs as 50. That is to say, we will randomly select 2% samples from the dataset to initialize the task model. Then, we query 2% extra unlabeled samples in each iteration for labeling. The process is iterated until the labeled set size reaches 20%. For the rest of the datasets, we set the start point nStart as 10%, the total budget nEnd as 70%, the step size nQuery as 5%, and the training epochs as 150. We use SGD as the optimizer since it performs better than Adam (Beck et al., 2021). The learning rate is set to 0.1 for all datasets. According to (Beck et al., 2021), resetting the model after each iteration achieve better performance than fine-tuning the model from previous rounds. Hence, we reset the model for each DAL iteration and retrain the model from scratch.

Table 2: The dataset and the fully-supervised model accuracy.

| Dataset | # Classes | Train/Test | Acc (%) |
|---------|-----------|------------|---------|
| MNIST | 10 | 60,000/10,000 | 97.68 |
| CIFAR-10 | 10 | 50,000/10,000 | 92.15 |
| GTSRB | 43 | 39,209/12,630 | 99.33 |

Please note that we evaluate each method for five runs, and we try to fix the five random seeds so that every method has the same initial accuracy. However, there are some exceptions: WAAL and LearningLoss. This is because WAAL uses unlabeled data for training, and the model learned by learning loss will be affected by the loss model.

**Platform and Hardware.** We use the Pytorch framework and two Titian V GPUs for evaluation.

**Evaluation metrics.** In this paper, we use three metrics to evaluate different DAL methods: *Accuracy*, *Average Performance Improvement (API)*, and *Cost*. The accuracy of a trained model is defined as the ratio between the correctly classified test inputs out of total test inputs. Due to the large number of methods we study, plotting the accuracy for all methods at each iteration in a single figure is hard for visualization. Hence, we use API, i.e., the average performance of a DAL method over all iterations. API is calculated as follows:

$$API_{al} = \frac{1}{N} \sum_{i=1}^{N} (acc_i^{al} - acc_i^{random}), \tag{1}$$

where $N$ represents the total iterations, i.e., $N = 1 + int(\frac{nEnd - nStart}{nStep})$. Also, the $acc_i^{al}$ stands for the accuracy achieved by a deep active learning method (e.g., CoreSet) at iteration $i$. The $acc_i^{random}$ stands for the accuracy achieved by RandomSampling at iteration $i$.

Besides, we also report the cost of each deep active learning method as the running time in seconds. The cost mainly contains two parts: the data selection part and the training part. The calculation of the cost is as follows:

$$Cost_{al} = \frac{1}{N} \sum_{i=1}^{N} (selection\_cost_i^{al} + training\_cost_i^{al}), \tag{2}$$

where $N$ represents the total iterations. The $selection\_cost_i^{al}$ and $training\_cost_i^{al}$ represents the selection cost and the training cost for certain active learning method at iteration $i$.

## 4 EXPERIMENTAL RESULTS AND ANALYSIS

### 4.1 PERFORMANCE COMPARISON

We report the performance of each method in Table 3. We break the results into the SAL part and the SSAL part. For ease of comparison, we report the average accuracy improvement over **RandomSelection**. Besides, we show the detailed accuracy vs. budget curve of these techniques in the Appendix (see Figure 7 in Appendix). We have several observations from the results.

**Comparison among different SAL methods.** We observe that RandomSampling is a strong baseline for SAL methods. For each dataset, many SAL methods perform worse than RandomSampling according to Table 3. To be specific, there are 6/14, 3/14, and 6/14 methods perform worse than RandomSampling for MNIST, CIFAR-10, and GTSRB, respectively. Also, most methods perform worse than RandomSampling in at least one dataset. Only BadgeSampling, ClusterMarginSampling, and LeastConfidence constantly perform better than RandomSampling on all studied datasets.

Another important observation for SAL methods is that: there is no SAL method that can outperform others on every dataset. For instance, out of the three methods that are better than RandomSelection, BadgeSampling outperforms the other two methods on MNIST, but it performs worse than LeastConfidence on GTSRB and CIFAR-10. More studies on SAL methods can be found in Appendix.

Table 3: The performance and cost comparison of different DAL methods. The performance is calculated as the **average accuracy improvement (API)** over RandomSampling. We bold the best method from both SAL and SSAL methods. Also, we highlight the rows that consistently outperform the RandomSampling. The cost is normalized using the cost of RandomSampling.

| | Performance | | | Cost | | |
|---|---|---|---|---|---|---|
| Strategy | MNIST | CIFAR-10 | GTSRB | MNIST | CIFAR-10 | GTSRB |
| The SAL methods | | | | | | |
| **RandomSampling** | 0.00 | 0.00 | 0.00 | 1.00 | 1.00 | 1.00 |
| **ActiveLearningByLearning** (Mayer & Timofte, 2020) | -0.15 | 1.58 | 0.34 | 1.51 | 0.86 | 1.04 |
| **AdversarialBIM** (Ducoffe & Precioso, 2018) | 1.44 | 0.25 | -0.24 | 6.39 | 1.49 | 2.27 |
| **BALDDropout** (Gal et al.) | 1.18 | 1.48 | -0.04 | 1.35 | 1.05 | 1.02 |
| **BadgeSampling** (Ash et al., 2020) | **1.84** | 1.89 | 0.27 | 2.74 | 1.52 | 2.08 |
| **ClusterMarginSampling** (Citovsky et al., 2021) | 1.76 | 0.28 | 0.23 | 1.44 | 0.87 | 1.08 |
| **CoreSet** (Sener & Savarese, 2018) | -1.15 | 0.95 | 0.26 | 1.69 | 1.17 | 1.08 |
| **KMeansSampling** (Zhdanov, 2019) | -0.22 | 0.17 | 0.20 | 5.62 | 1.82 | 1.40 |
| **LearningLoss** (Yoo & Kweon, 2019) | -2.32 | -2.38 | -0.94 | 0.66 | 0.62 | 0.75 |
| **LeastConfidence** (Wang & Shang, 2014) | 1.40 | **1.95** | **0.34** | 1.06 | 0.99 | 1.00 |
| **MCADL** (Yuan et al., 2019) | 0.57 | -0.50 | 0.03 | 7.21 | 1.41 | 1.32 |
| **CoreGCN** (Caramalau et al., 2021) | 1.21 | 1.16 | -8.52 | 2.35 | 1.25 | 1.12 |
| **UncertainGCN** (Caramalau et al., 2021) | -0.85 | 0.14 | -8.77 | 1.75 | 1.21 | 1.07 |
| **VAAL** (Sinha et al., 2019) | -0.34 | -0.49 | -24.79 | 3.03 | 6.62 | 6.57 |
| The SSAL methods | | | | | | |
| **WAAL** (Shui et al., 2020) | 3.00 | 2.05 | 0.21 | 9.18 | 3.90 | 1.78 |
| **SSLConsistency** (Gao et al.) | 3.39 | **5.83** | **1.19** | 12.11 | 4.82 | 3.03 |
| **SSLDiff2AugDirect** (Song et al., 2019) | 3.49 | 5.78 | **1.19** | 11.90 | 5.05 | 2.99 |
| **SSLDiff2AugKmeans** (Song et al., 2019) | **3.50** | 5.74 | 1.17 | 12.11 | 4.82 | 5.26 |
| **SSLRandom** | 1.68 | 4.57 | 1.11 | 11.70 | 5.76 | 4.93 |

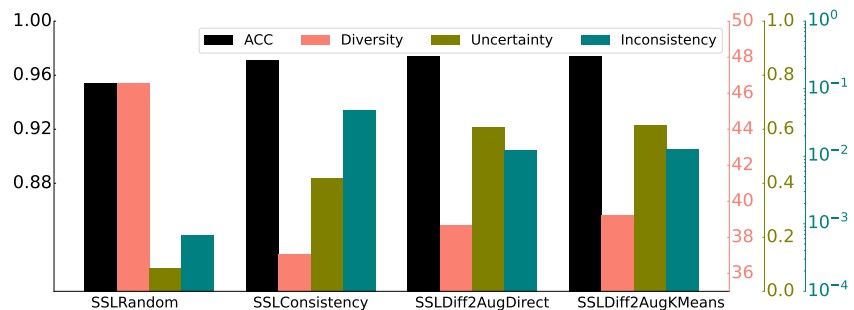

Figure 2: Comparison between different SSL-based active learning methods. These four methods use the same SSL method to train the model. This figure is produced using MNIST dataset.

**Comparison between SAL and SSAL methods.** When comparing SSAL with SAL, we have the following observations. First, in general, SSAL methods outperforms SAL methods for a large margin. On the one hand, the SSLRandom method has a 1.68, 4.57, and 1.11 average accuracy improvement over the RandomSampling on MNIST, CIFAR-10, and GTSRB, respectively. On the other hand, the SSAL improves the best method in SAL by 3.5-1.84=1.66, 5.83-1.95=3.88, 1.19-0.34=0.85 w.r.t. the API on MNIST, CIFAR-10, and GTSRB respectively. Second, SSAL methods are more stable than SAL methods. Each SSAL method outperforms the RandomSampling consistently on all datasets.

This indicates that using unlabeled data to train the task model is useful. The unlabeled data provides extra information for the model to learn so that the ability of the model can be improved.

**Comparison among different SSAL methods.** Within the SSAL methods, we observe that SSLConsistency, SSLDiff2AugDirect, and SSLDiff2AugKmeans are much better than SSLRandom, while WAAL sometimes performs worse than SSLRandom. We also observe that the performance difference between SSLConsistency, SSLDiff2AugDirect, and SSLDiff2AugKmeans are negligibly small.

We investigate the reason behind the above observations by studying the property of the selected data in Figure 2 and Figure 3. To generate these two figures, we randomly label 2% samples and

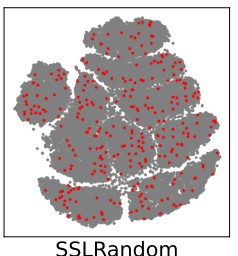 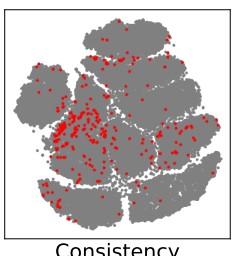 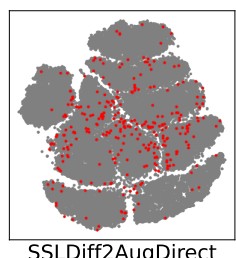 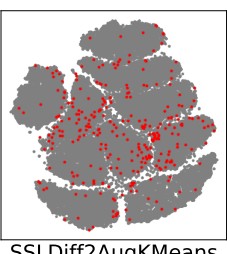

| SSLRandom | Consistency | SSLDiff2AugDirect | SSLDiff2AugKMeans |

Figure 3: Visualization of selected samples (red points) out of total samples (gray) for different SSAL methods. This figure is produced using the MNIST dataset.

train a model using SSL methods. Then, we select and label 1% extra samples to retrain the model and report the resulting accuracy. To simplify the process, we only experiment on SSLRandom, SSLConsistency, SSLDiff2AugDirect, and SSLDiff2AugKmeans. This is because they use similar semi-supervised training methods, and their difference largely comes from the selection strategies. Figure 2 shows the diversity, uncertainty, and consistency of the selected samples at certain iteration. The diversity of the selected samples is calculated as the average distance between selected samples. The uncertainty is calculated as $1 - prob$, where $prob$ stands for the highest confidence for one sample. The inconsistency is calculated as the prediction variations between one sample and its augmented version. Figure 3 visualizes the selected samples among all samples.

First, we find that choosing uncertain or inconsistent samples in the SSAL setting can improve the model performance consistently and significantly. This is why the performance difference between SSLRandom and the other three methods (e.g., SSLConsistency, SSLDiff2AugDirect, and SSLDiff2AugKmeans) is large. To illustrate, Figure 2 and Figure 3 shows that samples chosen by SSLRandom are much more certain and consistent. In contrast, the rest three methods focus more on uncertain or inconsistent samples. Remember that in the SAL setting, most selection methods perform worse than RandomSampling. This may be because semi-supervised learning provides the model with better feature extraction ability. Hence, they can estimate the uncertain samples more accurately than in the SAL setting.

Second, sample uncertainty is more important than sample diversity during sample selection in SSAL. Since SSLConsistency, SSLDiff2AugDirect, and SSLDiff2AugKmeans all focus on selecting uncertain samples, the performance differences between them are small. To illustrate, although SSLDiff2AugKmeans considers diversity during sample selection, the performance gain is tiny on MNIST, and there is no performance gain on CIFAR-10 and GTSRB. This result indicates that sample diversity is not as useful as sample uncertainty in the SSAL setting. The underlying reason may be that the learning procedure has already utilized all the unlabeled data and learned the diverse information in these data. Hence, focusing on uncertain samples is more urgent.

Hence, we conclude that active sample selection can significantly improve performance under the SSL setting. That is to say, SSAL is helpful for model performance improvement. Second, we prefer uncertain samples to diverse samples when applying SSAL methods since uncertain samples can bring better results.

## 4.2 COST COMPARISON

The cost of each method is shown on the right-hand side of Table 3. We normalize the cost of each method w.r.t. the random selection method. The original time cost for RandomSampling on MNIST, CIFAR-10, and GTSRB are 131, 1047, and 658 seconds, respectively. We have the following observations. First, SSAL methods introduce much more time cost than SAL methods. For example, SSAL methods consume around 4x to 6x energy than SAL methods on MNIST. This is because the semi-supervised training will evaluate every unlabeled and labeled sample to calculate the loss. This also means that when the number of unlabeled data increases, the training time will increase. For example, the increase of cost on MNIST is more than that on GTSRB because the number of unlabeled samples in MNIST is more than that in GTSRB.

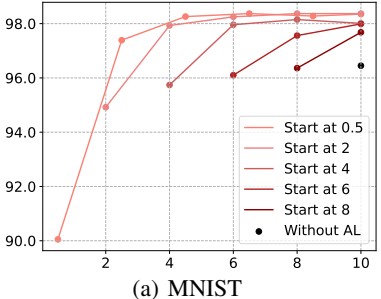
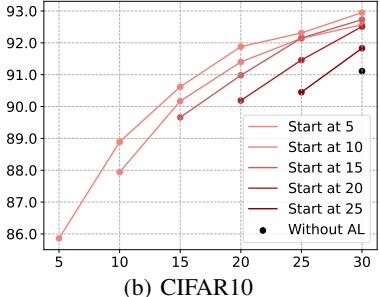

(a) MNIST  (b) CIFAR10

Figure 4: The accuracy achieved by SSLDiff2AugDirect when using different initial budgets (%). The x-axis represents the start budget measured in percentage, and the y-axis stands for the accuracy. We observe that the performance is negatively correlated with the initial budget. This indicates that we should apply the SSAL method as early as possible to achieve better performance.

## 5  IN-DEPTH INVESTIGATION OF SSAL ALGORITHMS

According to Section 4, SSAL is the most promising DAL method as it achieves the best performance across multiple datasets. In this section, we aim to conduct an in-depth analysis of the SSAL method and seek to provide practical guidance to users.

### 5.1  START SSAL AS EARLY AS POSSIBLE

The previous session shows the effectiveness of the SSAL methods. In this part, we investigate when to start the SSAL process would be the best, i.e., we want to decide a proper nStart for a given nEnd. To this end, we fix the total labeling budget and start the SSAL process at different points. Specifically, we set the total budget as 10% and 30% for MNIST and CIFAR-10, respectively. Since the SSAL methods have similar performance results, we randomly choose one, e.g., SSLDiff2AugDirect, to generate the result. The result is shown in Figure 4.

On the one hand, intuitively, starting the SSAL too late would result in poor performance. The later the SSAL process starts, the more samples are randomly sampled, and the accuracy is closer to SSLRandom. This can be validated by Figure 4. For example, we get around 2 percent accuracy loss if we start from 10% instead of 2% on MNIST. This accuracy loss is enormous. On the other hand, some may argue that starting the SSAL process too early will also cause problems. Because when the labeled data is highly sparse, SSAL can perform worse than SSLRandom (Mittal et al., 2019). To consider this, we start the process from 0.5% for MNIST, and the result shows that the final accuracy achieved by this starting point is the highest. Hence, we suggest starting the SSAL process as early as possible to achieve the best accuracy.

### 5.2  THE MORE UNLABELED DATA, THE BETTER

From Table 3, we observe that the effectiveness of SSAL methods is different for different datasets. We use the performance difference between a specific SSAL method and SSLRandom to reflect the effectiveness of that SSAL method. For example, the performance of SSLDiff2AugDirect over SSLRandom are 3.49/1.68 = 2.07, 5.78/4.57=1.26, 1.19/1.11=1.07, and 6.33/5.97=1.06 for MNIST, CIFAR-10, GTSRB, and ImageNet, respectively. We hypothesize that the performance improvement of SSAL methods is correlated to the number of unlabeled samples available for selection. As shown in Table 5.2, the average number of samples are 6000, 5000, 911, and 500 for MNIST, CIFAR10, GTSRB, and Tiny-ImageNet, respectively. More average samples per class mean more data is available for selection, in which case we have a higher probability of obtaining valuable samples.

To validate this hypothesis, we study the effectiveness of SSAL methods by fixing the dataset and only varying the amount of unlabeled data. Specifically, we initialize the model with 5000 labeled images randomly drawn from the CIFAR10 dataset. Then, we provide the model with different amounts of unlabeled data and use SSLDiff2AugDirect for model training and sample selection. The result is shown in Figure 5.2. The x-axis of Figure 5.2 stands for the percentage of data selected

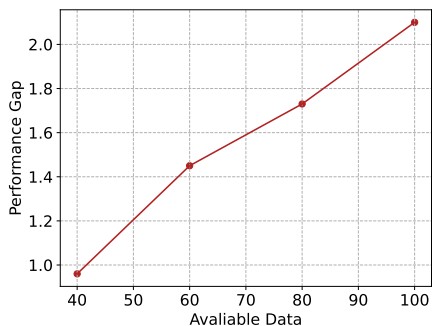

| Dataset | Average Training Sample Per Class | Performance Improvement |
|---------|----------------------------------|------------------------|
| MNIST | 6000 | 2.07 |
| CIFAR-10 | 5000 | 1.26 |
| GTSRB | 911 | 1.07 |
| Tiny-ImageNet | 500 | 1.06 |

Table 4: The average training sample per class and the performance improvement for different datasets.

Figure 5: The performance gap between SSLDiff2AugDirect and SSLRandom w.r.t the number of unlabeled data on CIFAR10 dataset.

from the entire CIFAR10 dataset to train the model. For example, if the selected portion is 40%, then the amount of unlabeled data equals to 40% × 50000 - 5000 = 15000 images. We observe that the performance gap between SSLDiff2AugDirect and SSLRandom increases with the increase of unlabeled data. This result indicates that active data selection performs better if more unlabeled data is available. Hence, we recommend the user collect more unlabeled data before performing SSAL.

### 5.3 THE QUERY SIZE HAS LITTLE INFLUENCE ON SSAL PERFORMANCE

We show the influence of the query size on SSAL methods in Figure 6. Intuitively, the query size can influence the performance of the SSAL methods. If the query size is too small, the selected samples are too less to be statistically significant. If the query size is too big, the selection would be coarse-grained. In this experiment, we fix the starting budget as 10% and the total budget as 50%. Then, we train the model by varying the query size in each iteration. Despite the performance of SLDiff2AugDirect and SSLConsistency decreasing slightly, their performance is still much higher than SSLRandom. Hence, the SSAL methods can perform well in various query sizes.

Figure 6: The influence of query size on the performance of SSAL methods.

### 6 CONCLUSION

This paper studies the effectiveness of existing deep active learning methods. Our study is the most comprehensive one, which covers 19 DAL methods, including the state-of-the-art SAL and SSAL ones. Through extensive experiments, we observe that the emerging SSAL techniques provide promising results. Specifically, the traditional SAL methods can hardly beat random selection, and no SAL method can consistently outperform others. However, the SSLAL methods can easily surpass all SAL methods, and we find that active sample selection in SSAL can bring huge performance improvement. In this paper, we also conduct an in-depth analysis of the SSAL methods and give two guidances to the practitioners. First, one should perform SSAL as early as possible and seek more unlabeled data whenever possible to achieve better performance.

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

# A APPENDIX

## A.1 EXPERIMENTAL SETTINGS OF DIFFERENT DAL PAPER

We summarize the detailed settings of some representative DAL papers in Table 5. As we can observe, these hyperparameters differ a lot in different papers, including the start budget size (Start), step size, learning rate (LR), training epoch, accuracy calculation criteria, and optimizer. This is the reason why we cannot compare their performance only by looking for the reported numbers in their paper. Our empirical study unifies these settings and provides a fair comparison of these methods.

Table 5: Hyperparameters used by different active learning methods.

| Method | Start | Step Size | LR | Training Epoch | Accuracy Calculation | Optimizer |
|---|---|---|---|---|---|---|
| BadgeSampling | 0.2% | 0.2%/2%/20% | 1e-3 | Terminate when 99% accuracy archived | After train | adam |
| ClusterMarginSampling | 10% | 10% | 1e-3 | - | - | batch SGD |
| CoreSet | 10% | 10% | 1e-3 | - | | RMSProp |
| KMeansSampling | 10% | 10% | 1e-3 | - | | RMSProp |
| LearningLoss | 2% | 2% | 1e-1, 1e-2 | 200 | After train | SGD |
| MCADL | 4% | 128 | - | - | - | Adam |
| WAAL | 2% | 2% | 1e-2 | 80 | After train | SGD |
| CoreGCN/UncertainGCN | 2% | 2% | 1e-1, 1e-2 | 200 | - | SGD |
| Ensemble | 0.4% | 0.8% | - | 150 | - | RMSprop |
| VAAL | 10% | 5% | 1e-2 | 100 | Use best model on validate set | SGD |

## A.2 DETAILED PERFORMANCE OF DAL METHODS

We show the detailed accuracy against the budget curve in Figure 7. Note that we also include the performance of Tiny-ImageNet to privide more information. Tiny-ImageNet is a subset of the ILSVRC- 2012 classification dataset. It consists of 200 object classes. All images were down-sampled to 64 × 64 × 3 pixels. Tiny-ImageNet is a more practical and complex dataset. As we can observe, the SSAL methods consistently outperform SAL methods in all datasets.

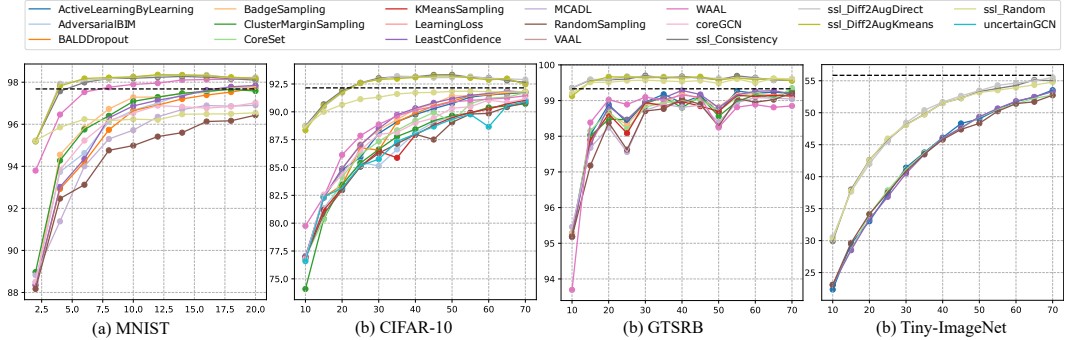

Figure 7: The performance of the deep active learning techniques. Note that for better visualization, we only draw the lines for methods that is better than random selection. The x-axis denotes the budget in percentage, and the y-axis denotes the accuracy.

## A.3 ROBUSTNESS STUDY OF SAL METHODS

We investigate two factors that can have a large influence on the performance of the SAL method: the initial budget and the query size. If the initial budget is too small, the initial model can be poorly trained. Selecting new data using this model can cause less optimal results. If the initial budget is too large, most of the labeled samples are randomly selected, and hence the model accuracy would be similar to that of random selection. If the query size is too small, the selected samples are too less to be statistically significant. If the query size is too big, the selection would be coarse-grained. To study these two factors, we choose five representative methods: BadgeSampling, LeastConfidence, CoreGCN, ClusterMargin, and BALDDropout. The experiment in this section is performed on CIFAR-10.

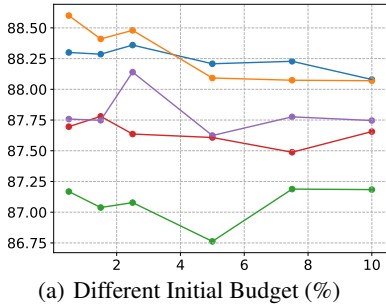 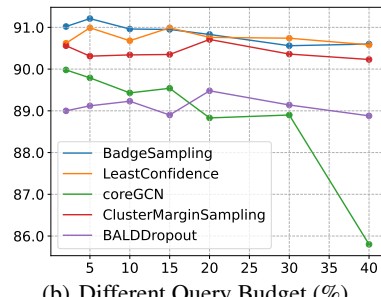

(a) Different Initial Budget (%)       (b) Different Query Budget (%)

Figure 8: The performance of five SAL methods under different initial budget and query budget. (a) the impact of the initial budget. The nQuery and nEnd are 5% and 40%, respectively. (b) the impact of the query budget. The nStart and nEnd are 10% and 50%, respectively.

**Start budget.** The influence of the initial budget on the performance is shown in Figure 8 (a). In this experiment, we fix the total budget as 30% and the query size as 2% for each method, and train the model with different initial budgets. From Figure 8 (a), we observe that the performance of each method does not change much with different initial budgets.

**Query size.** The influence of the query size in each DAL iteration is shown in Figure 8 (b). In this experiment, we fix the initial budget as 10% and the end budget as 50%. Then, we train the model by varying the query size in each iteration. We observe that the performance decreases when we increase the query size. This is because the redundancy can increase when we select more samples at once. Note that our observation is in contrast with Beck et al. (2021), where the authors claims that the query size has little influence on the performance. We suspect the difference is that the query size in Beck et al. (2021) is too small to reflect the trend.

