# OpenReview forum: "An Empirical Study on the Efficacy of Deep Active Learning Techniques"
_ICLR.cc/2023/Conference — Submitted to ICLR 2023_

### Official Review · Reviewer_ysmb · 2022-10-17

**Confidence:** 4
**Correctness:** 3
**Technical Novelty And Significance:** 2
**Empirical Novelty And Significance:** 3
**Recommendation:** 6

**Clarity, Quality, Novelty And Reproducibility:**

### Clarity
The main idea of this paper is quite straightforward. The experimental setup and the key results are easy to understand.
### Quality
This is the only caveat here - while the paper examines a good set of approaches using the correct metrics, the conclusions are still based on (too) academic datasets.
### Reproducibility
I think most of the approaches that are tested here are easy to implement, and I even found a few open source version for some. In addition, the authors promise to make the code publicly available.
### Novelty
The novelty of this work, as in most benchmarks, is very limited. The main goal here, as I see it, is to shed some light on the existing state of the art rather than pushing it forward

**Strength And Weaknesses:**

## Strength
This paper is well written and easy to follow.
Most recent DAL methods are tested here, with the important emphasis both on (pre-) processing of unlabeled data, as well as the cost of such an approach.

The main insight of this work seems important - DAL struggles to beat the ransom baseline without usage of such unsupervised approaches.

## Weakness
- The one notion that weakens this work is the specific datasets on which this benchmark is based, in two aspects:
   1) There are substantial evidence that both MNIST and CIFAR are saturated (= contain non-negligible label noise)y, but mainly do not pose challenges like class-imbalance or long-tail distribution, which is the one of the main goals of DAL, IMHO.
   2) Why did the authors not tackle larger datasets - while there are some initial works using the (old but very large) ImageNet, I would argue that there are quite a few examples between the latter and (the too academic academic) MNIST/CIFAR
- An important question here is regarding the **re-implementation** of all methods - some of the methods have made the code publically available, and such re-implementation might be unfair (both in performance and cost). I would expect, at the very least, to compare to the original implementation when available - or to state formally that there is little difference in performance or cost.
- The authors state that **5 runs were made** for each approach, why is the STD not reported?
- I understand that 2%/5% increments are from the previous benchmarks, but why not test smaller increments?
- SSLRandom is not properly defined in the text.
- I would strongly recommend using the term *self-supervised* whenever appropriate - that is whenever an augmented/corrupted version of the data is compared to the original (or another augmented version)

**Summary Of The Paper:**

This paper performs a benchmark of active learning for deep nets (aka DAL).
The authors make the distinction between _supervised_ methods and _semi-supervised_ methods (respectively SAL and SSAL) by the ability to harness unlabeled data.
The proposed benchmark is comprehensive (as in the amount of methods tested) and puts an emphasis on both on usage of unlabeled data as well as measuring the cost of the *selection* process.

The supervised flavor of DAL is further divided into three types of data selection:
- Uncertainty based selection - uses the model's output to estimate the uncertainty a data sample, and select the ones with closest to the decision boundary.
- Diversity/Representative-ness based selection - selection that maximizes the diversity of the data points, usually over a set of clusters or a metric space.
- Hybrid selection - a combination of the two former approaches

Semi-supervised approaches perform some of the learning process without the labels.
Three of the methods here use an approach termed in the literature as *self-supervised*, which are based on comparison of two forward runs through the net, one of which is of an augmented/corrupted version of the data. A fourth approach, WAAL, is base on a adversarial approach of training a discriminator-net to differentiate between the labeled an unlabeled data.

---

One of the main findings of this work is that unlabeled data should be mapped (as soon a possible), by and unsupervised approach.
Methods that do not harness such an un/semi- supervised approach might not achieve performance better that random selection

**Summary Of The Review:**

This work provides an empirical evidence of how DAL should be addressed in practical sense. The notion that unlabeled data should *preferably* be consumed *earlier* via unsupervised methods is not surprising, but important - and was not addressed in most prior benchmarks.

---
The main limitation of this work is that it is based on academic/toy datasets like MNIST and CIFAR, as I detail above.


---

Nevertheless, this work passes the bar IMHO, and lay a basis for future works to complete the gap(s) I mentioned

---

> ### Author Response · Authors · 2022-11-19
> **To Reviewer #ysmb**
>
> ### **Please see the revised paper at this anonymous link: https://we.tl/t-lbn4R5O90d. The code is at: https://we.tl/t-IFUwsjrPsy**.
>
> **Q1: The main limitation of this work is that it is based on academic/toy datasets like MNIST and CIFAR.**
>
> Please refer to the common question CQ1.
>
> **Q2: I would expect, at the very least, to compare to the original implementation when available - or to state formally that there is little difference in performance or cost.**
>
> For the open-sourced methods, we try to maximumly reuse their original code.  We do not change their sample selection function and directly apply their function to our DAL pipeline. In this way, we can fairly compare the performance and the cost of different methods.
>
> **Q3:  The authors state that 5 runs were made for each approach, why is the STD not reported?**
>
> Our previous version only reports the mean performance due to space limitations.  We shall revise our paper to include this information in our next version.
>
> **Q4: I understand that 2%/5% increments are from the previous benchmarks, but why not test smaller increments?**
>
> We have tested a wide range of query sizes in our paper. For SSAL methods, we vary the query size from 2.5% to 20% for CIFAR-10 and observe the accuracy difference. The result is shown in Figure 6, and it indicates that SSAL methods perform well in various query sizes. For SAL methods, we vary the query size from 2.5% to 40%, and the result is shown in Appendix-Figure 8 (b). This result indicates that if the query size is too large, the final accuracy would be reduced significantly. Please refer to the corresponding section for more information.
>
> **Q5：SSLRandom is not properly defined in the text.**
>
> There are two key steps in the DAL pipeline: the sample selection step and the model training step. SSLRandom selects samples randomly in the sample selection step and uses a semi-supervised training method (e.g., MixMatch) in the model training step. We have modified our paper to give the definition of SSLRandom (See Section 2.2).
>
> **Q6: I would strongly recommend using the term *self-supervised* whenever appropriate - that is whenever an augmented/corrupted version of the data is compared to the original (or another augmented version)**
>
> Semi-supervised learning is different from self-supervised learning. In semi-supervised learning, part of the samples has labels and some samples are unlabeled. The goal is to make use of both the labeled and unlabeled data. In contrast, self-supervised learning means the supervision signal comes from the data itself, and it can work without any labeled data. Self-supervised techniques can be used by semi-supervised learning methods to make use of unlabeled data [1].
>
> [1] Zhai, X., Oliver, A., Kolesnikov, A., & Beyer, L. (2019). S4L: Self-Supervised Semi-Supervised Learning. *2019 IEEE/CVF International Conference on Computer Vision (ICCV)*, 1476-1485.

---

### Official Review · Reviewer_9BF2 · 2022-10-20

**Confidence:** 3
**Correctness:** 3
**Technical Novelty And Significance:** 3
**Empirical Novelty And Significance:** 3
**Recommendation:** 5

**Clarity, Quality, Novelty And Reproducibility:**

The paper is clear, with good novelty. The quality of the paper is overall high.

However, the paper would need to publish full code base details to have good reproducibility, also, as I mentioned in the weakness part, the model architecture detail is a key aspect of reproducibility as well.

**Strength And Weaknesses:**

Strength:

The drawing of figure 1 is really cute. The related work is well done with enough details and references. The clarity of the paper is strong and evaluation metrics are nicely introduced.

The scope of the paper is a strength with a focus on benchmarking DALs together in a similar setting. Directly exposing the weakness of current DAL approaches that they sometimes don't even beat random sampling (the majority of them from Table 3).


Weaknesses:

The title is broader than the study. Wil active learning is a more and more broad term used by more communities, since only image classification tasks are studied in this work, I suggest adding such limitation in the title.

The comparison feels unfair frankly speaking. As this work aims to evaluate the efficacy of "ACTIVE LEARNING", comparing semi-supervised AL with random sampling is not fair. Instead, authors should consider comparing SSAL with semi-supervised learning to show the effect of active learning piece.

The input dimension of the datasets are too small (28 x 28) for image classification tasks from my perspective. As majority of real-life applications would be carried out in larger image scales, whether the same conclusion holds for larger image sizes is a gap for me and impacts the significance of this work.

Claim 5.1 "START SSAL AS EARLY AS POSSIBLE": The smallest starting point in the experimental result is 0.5% instead of 0%. With minimal starting labels, the models would not even be appropriately trained, and therefore I highly doubt the efficacy of such a model providing good proposals for unlabeled points to be labeled. I understand the authors' suggestion, but the claim needs to be modified as this is not true (at least not proved if you start from 0.5%, not 0%).

What model architecture was used across the different tasks? This seems like a crucial detail that should be included in the experimental setting.

For the cost in the unit of seconds, I am wondering why some active learning methods can be faster than random sampling?



Questions:
1. For such a comprehensive analysis, the exclusion of ImageNet seems odd to me, if the computational resource is an issue, smaller version of ImageNet can be considered?
2. Why is the step size of MNIST different from the other two methods?
3. May I request the proof of "SGD optimizer performs better than Adam" in the supplementary? This seems to be controversial that a single citation is not enough.
4. Did the learning rate have any decay scheduler? Active learning should mimic the real process of the machine learning cycle and therefore, the training process should be mimicking the same effort of a practitioner training for the best performance.
5. What does Figure 3 convey? I didn't see the contribution of supporting any claims from Figure 3.

**Summary Of The Paper:**

This is an empirical study of the efficacy of DAL on image classification tasks. It benchmarked 19 DAL algorithms, grouped into supervised and semi-supervised and tested on three simple different datasets (MNIST, CIFAR, GTSRB). After benchmarking, it concluded that the semi-supervised active learning techniques does a much better job than SAL and traditional SAL is not always better than even random sampling.

**Summary Of The Review:**

I think the paper definitely has its contribution to the field, pointing out the problem in the AL field that they might not work as advertised when benchmarked carefully together and that SSAL performs great. However, as I mentioned in the weakness part, there are several important misses in the current form of the argument that needs to be addressed until this gets published. Therefore I would like to recommend reject/marginally reject of its current form, with a open mindset to change my ratings once those concerns are addressed.

---

> ### Author Response · Authors · 2022-11-19
> **To Reviewer 9BF2**
>
> ### **Revised paper link: https://we.tl/t-lbn4R5O90d. Code: https://we.tl/t-IFUwsjrPsy**.
>
> **Q1: Only image classification tasks are studied.**
>
> We have modified our title to restrict our scope on the image classification task.
>
> **Q2: Comparing semi-supervised AL with random sampling is not fair.**
>
> There are two main components in the active learning pipeline:  sample selection and model training. Both SAL and SSAL can use random sampling for the sample selection, the difference between them is whether they use unlabeled samples during model training. Comparison between the SSAL method and SSL-based random sampling, i.e., SSLRandom, is fair because they use the same model training method, and the only difference between them is the sample selection process. Please note that SSAL and semi-supervised learning are not comparable because the former contains the sample selection and model training, while the latter is only a way for model training.
>
> **Q3: Whether the same conclusion holds for larger image sizes.**
>
> Please refer to the common question CQ1.
>
> **Q4: "START SSAL AS EARLY AS POSSIBLE".**
>
> We agree that starting the process too early (e.g., around 0%) may cause problems. For example, training the DNN model with several samples does not make sense. However, our empirical study shows that starting SSAL too late would be less optimal. That’s why we suggest starting SSAL early to maximally utilize its potential. Based on these, we would revise our statement as follows: once the randomly selected samples can make the training process go well, we can consider starting the SSAL process. We have revised our paper accordingly in Section 5.1 to make this clearer.
>
> **Q5:Model architecture**
>
> For MNIST, we use the LeNet architecture. For the rest datasets, we use the well-known ResNet-18 model. We have revised our paper accordingly to include this information (See Table 2).
>
> **Q6: Why some active learning methods can be faster than random sampling?**
>
> Since we include the model training time in cost calculation, the cost can be affected by training issues (e.g., GPU scheduling and the interaction between different training tasks).  Hence, the calculated cost has some noise. Since the cost difference between many SAL methods is small (e.g., RandomSampling and LeastConfidence), the noise can make the cost of some SAL methods less than the RandomSampling.
>
> As our goal is to study the cost difference between SAL and SSAL, we turn to calculate the average cost for SAL and SSAL methods and report the result in Table 4. Due to the large difference between the cost of SAL and SSAL, the noise cannot influence their relative relationship much.
>
> **Q7: Why is the step size of MNIST different from the other two methods?**
>
> This is because MNIST is a relatively simple task and needs fewer data to reach high accuracy. For example, the accuracy is already saturated when the portion of labeled samples reaches 20%. In this case, if we set the step size too large (e.g., 5%), then we can only observe the performance comparison of three iterations. Hence, we set the step size small to have more observation points to have a better comparison between different methods.
>
>  **Q8:  May I request the proof of "SGD optimizer performs better than Adam" ?**
>
> We adopt SGD because many DAL methods use it for training, and (Beck et al., 2021) say it is better.  However, studying which optimization method is better, i.e., Adam or SGD, is not the main focus of this paper. Our main goal is to compare different DAL methods under some fixed settings. Hence, adopting either SGD or Adam is acceptable for us.
>
> **Q9: Did the learning rate have any decay scheduler?**
>
> We agree that the learning rate decay influences the final accuracy. In our paper, we include two learning rate decay stages. In each stage, we decay the learning rate by 0.1. For MNIST, we set the first and the second decay stage at epochs 20 and 40, respectively. Since the rest datasets need more epochs to train the model, we set the decay stages at epochs 80 and 120, respectively. We revised our paper to include this information (See Section 3 — Training settings).
>
>  **Q10: Figure 3**
>
> Figure 3 serves as evidence for Figure 2 to show that the calculation of diversity and uncertainty is correct.
>
> Remember that we use a case study to show how different SSAL methods prefer different samples during sample selection. Figure 2 and Figure 3 are both used for this purpose. At a certain iteration on MNIST, Figure 2 gives some numeric properties (e.g., diversity and uncertainty) of the selected samples for each SSAL method, while Figure 3 visualizes the distribution of selected samples.
>
> For example, as we observe from Figure 2, the selected sample by SSLRandom is more diverse than that of other methods, and the distribution of selected samples in Figure 3 confirmed this.
>
> We revised our paper to provide this illustration (See Section 3 — Comparison among different SSAL methods).

---

### Official Review · Reviewer_aA7n · 2022-10-24

**Confidence:** 4
**Correctness:** 3
**Technical Novelty And Significance:** 2
**Empirical Novelty And Significance:** 3
**Recommendation:** 3

**Clarity, Quality, Novelty And Reproducibility:**

Clarity: this paper is easy to follow.

Quality: the experimental analysis and the related work should be enriched as mentioned in previous part.

Novelty: it is an empirical study. The novelty analysis is not applicable here.

Reproducibility: the author does not provide code, it brings difficulty to check whether their implementations is right or not.

**Strength And Weaknesses:**

Strengths: this paper focus on the difference beween SAL and SSAL and conduct comparative experiments, it's quite interesting and meaningful. We can see from the comparative experiments that SSL does promote the AL performance.

Weakness: the evaluation of a empirical study is to see how many new things it brings to us and whether the experiments are reliable. However, this paper is somewhat limited:

1. About the comparison with existing empirical studies, the author should at least add: 1) [r1], conduct experiments on 19 DAL approaches across 10 datasets, the author said "Our study is the most comprehensive one, which covers 19 DAL methods,
including the state-of-the-art SAL and SSAL ones". But the experimental scale of this paper is smaller than [r1], [r1] also contain SSAL ones like WAAL; 2) [r2] DAL methods for NLP tasks; 3) [r3] empricial study of how AL affected by semi-supervised and self-supervised learning. Additionally, the author should provide a detailed analysis of comparison between the findings of their paper and related work like (Mittal et al., 2019) and [r3].

2. The author said "Another important observation for SAL methods is that: there is no SAL method that can outperform
others on every dataset". But in this empirical study, only 3 datasets are included, furthermore, MNIST is a very simple task and it can achieve more than 95% accuracy with a few data under strong basic classifiers, it's not enough representative. These datasets are just image classification tasks. For other tasks, e.g., in (Mittal et al., 2019), they found that random selection with SSL performs best under semantic segmentation tasks. The basic tasks should be enriched.

3. In Table 3, the performance of learningloss method is very strange. At least in [r1], the performance of LearningLoss is fairly good on CIFAR10. I cannot check the code since the author does not provide the code. Maybe there is because the learning rate is too high, as analysed in [r4] appendix E, high learning rate will lead poor performance.

4. In Table 3, why UncertainyGCN and CoreGCN differs so much?

5. The author said they run each method 5 runs, can author provide variance/standard deviation/error bars of the 5 runs?

6. GTSRB is a imbalanced dataset, but lack of analysis of how different AL methods perform on imbalance data. Does author have new findings?

[r1] Zhan X, Wang Q, Huang K, et al. A comparative survey of deep active learning[J]. arXiv preprint arXiv:2203.13450, 2022.

[r2] Dor L E, Halfon A, Gera A, et al. Active learning for BERT: an empirical study[C]//Proceedings of the 2020 Conference on Empirical Methods in Natural Language Processing (EMNLP). 2020: 7949-7962.

[r3] Chan Y C, Li M, Oymak S. On the Marginal Benefit of Active Learning: Does Self-Supervision Eat Its Cake?[C]//ICASSP 2021-2021 IEEE International Conference on Acoustics, Speech and Signal Processing (ICASSP). IEEE, 2021: 3455-3459.

[r4] Mohamadi M A, Bae W, Sutherland D J. Making Look-Ahead Active Learning Strategies Feasible with Neural Tangent Kernels[J]. arXiv preprint arXiv:2206.12569, 2022.

**Summary Of The Paper:**

This paper conducts an expirical study of deep active learning, considering 19 approaches on 4 datasets. For the choice of deep AL approaches, they consider both supervised learning and semi-supervised learning based approaches.


**Summary Of The Review:**

This paper provides an empirical study of how SSL boosts AL performance. This paper still needs to be improved by enriching the comparisons with related work, especially (Mittal et al., 2019) and [r3].

---

> ### Author Response · Authors · 2022-11-19
> **To Reviewer aA7n**
>
> ### **Please see the revised paper at this anonymous link: https://we.tl/t-lbn4R5O90d. The code is at: https://we.tl/t-IFUwsjrPsy**.
>
> **Q1: About the Comparison with existing empirical studies,the author should at least add: 1) [r1], the experimental scale of this paper is smaller than [r1]; 2) [r2] DAL methods for NLP tasks; 3) [r3] empricial study of how AL affected by semi-supervised and self-supervised learning. Additionally, the author should provide a detailed analysis of comparison between the findings of their paper and related work like (Mittal et al., 2019) and [r3].**
>
> We address the above concerns as follows:
>
> - For the first problem, although their work includes 19 methods, some of them are very similar. For example, KMeans and KMeans(GPU) are, in fact, one method with different realizations. Also, they only include one SSAL method WAAL, and thus they can only compare WAAL with other SAL methods. In contrast, we cover five SSAL methods. This enables us to analyze the difference within SSAL methods and draw some new conclusions: 1) strategic sample selection can significantly improve performance over random selection under the SSAL setting; 2)  sample diversity is not as important as sample uncertainty in the SSAL setting.
> - For the second problem, please refer to the common question CQ2.
> - For the third problem, [r3] claims that strategic sample selection is no better than random selection in the SSAL setting, while our observation contradicts theirs.  After carefully reading their paper, we find that they only use three sample selection methods (e.g., random, CoreSet, and VAAL) for validation and ignore the state-of-the-art SSAL methods (e.g., uncertainty-based ones), which makes their conclusions less convincing.
> - Mittal et al., 2019 also include SSAL methods in their evaluation. Our work differs from their work in several ways. First, they only focus on the comparison between SAL and SSAL methods. At the same time, our work also takes an in-depth study of the SSAL methods, including figuring out what’s essential in SSAL, the start point of the SSAL techniques, and the impact of the number of unlabeled samples. Second, they only study 10 methods in total, while we provide a more comprehensive study on 19 state-of-the-art methods. Moreover, they do not study the cost of each technique.
>
> Since it is relevant and officially published, we have included the additional reference [r3] in our paper (See Table 1). [r1] and [r4] are arXiv papers, and [r2] is not for the image classification task. Besides, we have added a detailed comparison between our work, (Mittal et al., 2019), and [r3] in the revised paper (See Section 2.3).
>
> **Q2: The basic tasks should be enriched.**
>
> Please refer to the common question CQ1.
>
> **Q3: The performance of learningloss method is very strange. Maybe there is because the learning rate is too high.**
>
> For the LearningLoss method, we set the learning rate = 0.1. This learning rate is used for all datasets and methods. We choose this learning rate because this learning rate can provide a better model in most cases. We set the learning rate to [0.05, 0.01, 0.1, 0.5] to test the impact of different learning rates on the learning loss method. The resulting accuracies on CIFAR-10 when labeled data reaches 70% are [88.3, 88.6, **88.8**, 80.4] in percentage. As we can see, the learning rate = 0.1 generates the best model, but the accuracy is still lower than the random selection method, whose accuracy is **91.1%**. Hence, we think the performance is due to the technique instead of the learning rate setting.
>
>
> **Q4: Why UncertainyGCN and CoreGCN differs so much?**
>
> Even though they are both based on the representation learned by a GCN, they use different approaches to select samples. CoreGCN only considers the sample diversity while UncertainGCN only considers the uncertainty during the sample selection. This difference makes the resulting accuracy very different.
>
> **Q5: Can author provide variance/standard deviation/error bars of the 5 runs?**
>
> We only report the mean performance due to space limitations.  We shall revise our paper to include this information in our next version.
>
> **Q6: GTSRB is a imbalanced dataset, but lack of analysis of how different AL methods perform on imbalance data. Does author have new findings?**
>
> Generally speaking, as we can observe from Table 3, GTSRB prefers SAL methods that consider diversity over the ones that consider uncertainty. This is easy to understand, since the dataset is imbalanced, selecting diverse samples can make the labeled dataset more balanced and hence generate a better model. Otherwise, the selected dataset can be highly imbalanced and result in a poor model.

---

> > ### Comment · Reviewer_aA7n · 2022-12-04
> > **New paper and code.**
> >
> > I can't get access to the new paper and code through the provided links.

---

> > > ### Author Response · Authors · 2022-12-04
> > > **Paper and Code**
> > >
> > > After you click the above link, you need to agree the terms and services, and then you can see the "download" button to download the files.

---

### Official Review · Reviewer_5Vo7 · 2022-10-25

**Confidence:** 3
**Correctness:** 3
**Technical Novelty And Significance:** 2
**Empirical Novelty And Significance:** 3
**Recommendation:** 5

**Clarity, Quality, Novelty And Reproducibility:**

- I did not find the link to code or zipped code in the submission. I missed it but if it is not there, this would considerably reduce the merit of the work. Especially, in the context of a survey, having the code well-documented, reproducible and publicly available to rerun experiments, and extend to other methods easily would be a strength.

**Strength And Weaknesses:**

+ The paper is clearly written.
+ Several relevant methods have been compared and key insights clearly highlighted.

- The datasets are toy datasets
- I would have been very interested in seeing this evaluation in the context of NLP (where data augmentation has limited benefits compared to vision) and AL/HITL is a key component of modern ML systems.

- Page 7 paragraph 3, "Second, sample uncertainty is more important than sample diversity during sample selection in
SSAL." I would have liked to see some empirical evaluation highlighting this - the explanation provided in the text is a bit unclear to me (or likely i misunderstood).

Minor corrections:

- Page 5, "Platform and Hardware". "Titian" -> "Titan"

**Summary Of The Paper:**

The authors review 19 methods for supervised and semi-supervised active learning and benchmark their performance on three small CV datasets (MNIST, CIFAR, GTSRB).   The paper shows that compared to supervised active learning, a semi-supervised active learning (SSAL) approach is consistently better (with RandomSampling being a baseline which many of the SAL methods underperform) - Further, they show that SSAL with uncertainty objective improves over SSLRandom and that sample uncertainty is more important than diversity in the SSAL setting. Further, the query size has limited role and higher unlabelled data helps.

**Summary Of The Review:**

Overall it is a nicely written survey which makes its point clearly. This is good since there is considerable value to this kind of evaluation work since it gleans insight from existing works each focussed on individual approaches.  However, in terms of contribution, I would have liked to see (1) real-world datasets if focussing only on vision; and ideally, (2) similar analysis in another application domain to highlight wider applicability of the lessons gleaned.

---

> ### Author Response · Authors · 2022-11-19
> **To Reviewer 5Vo7**
>
> ### **Please see the revised paper at this anonymous link: https://we.tl/t-lbn4R5O90d**.
> **Q1: The datasets are toy datasets.**
>
> Please refer to the common question CQ1.
>
>
> **Q2: I would have been very interested in seeing this evaluation in the context of NLP (where data augmentation has limited benefits compared to vision) and AL/HITL is a key component of modern ML systems.**
>
> Please refer to the common question CQ2.
>
>
> **Q3: Page 7 paragraph 3, "Second, sample uncertainty is more important than sample diversity during sample selection in SSAL." I would have liked to see some empirical evaluation highlighting this - the explanation provided in the text is a bit unclear to me (or likely i misunderstood).**
>
> The above conclusion is obtained from our empirical results in Table 3 and Figure 2. We illustrate it as follows:
>
> - First, from the overall performance comparison in Table 3, we observe that **SSAL methods with diversity** (e.g., SSLDiff2AugKmeans) **can perform worse than the ones without diversity** (e.g., SSLDiff2AugDirect). In contrast, **the ones with uncertainty** (e.g., SSLConsistency, SSLDiff2AugDirect, and SSLDiff2AugKmeans) **perform better than the ones without** (e.g., SSLRandom).
> - Second, we also observe this phenomenon in the case study shown in Figure 2. Figure 2 shows the selected samples' diversity, uncertainty, consistency, and accuracy at a particular iteration. From this iteration, we observe that **the sample diversity does not have a clear relationship with the achieved accuracy, but the uncertainty does (choosing samples with higher uncertainty often achieves higher performance).**
>
> From these two empirical observations, we conclude that sample uncertainty is more important than sample diversity in the SSAL setting. The underlying reason could be that the SSAL learning process already uses diverse information in the unlabeled data. Once the uncertain ones are solved, the diverse information in the rest samples can be utilized naturally.
>
> We have revised our paper to clarify this (See Section 4.1).
>
> **Q4: I did not find the link to code or zipped code in the submission.**
>
> The code is at: https://we.tl/t-IFUwsjrPsy

---

### Author Response · Authors · 2022-11-19
**Common Questions**


### **Please see the revised paper at this anonymous link: https://we.tl/t-lbn4R5O90d. The code is at: https://we.tl/t-IFUwsjrPsy**.

### **CQ 1: The scale of the datasets.**

**We add more experiments on two larger datasets: CIFAR-100 and Tiny-ImageNet (See Table 3)**.  CIFAR-100 contains 60,000 real-world images of 100 classes.  The image size is 32 x 32. Tiny-ImageNet contains 110,000 images of 200 classes. The size of each image is 64 x 64. The results on these two datasets further support our claims.

With these two additional datasets, **our work is the most comprehensive compared to existing ones.** Specifically, Beck et al. use CIFAR-10, CIFAR-100, MNIST, Fashion-MNIST, and SVHN in their study [1]. Hu et al. use two image datasets, MNIST and CIFAR-10 [2]. Munjal et al. validate their results mainly on CIFAR-10 and CIFAR-100 and provide limited results (three SAL strategies) on ImageNet. Mittal et al. use CIFAR-10 and CIFAR-100 for validation. In contrast, we deliver results on MNIST, CIFAR-10, GTSRB, CIFAR-100, and TinyImageNet for many DAL methods.

[1] Nathan Beck, Durga Sivasubramanian, Apurva Dani, Ganesh Ramakrishnan, and Rishabh K. Iyer. Effective evaluation of deep active learning on image classification tasks. CoRR, abs/2106.15324, 2021.

[2] Qiang Hu, Yuejun Guo, Maxime Cordy, Xiaofei Xie, Wei Ma, Mike Papadakis, and Yves Le Traon.Towards exploring the limitations of active learning: An empirical study. In 36th IEEE/ACM International Conference on Automated Software Engineering, ASE 2021, Melbourne, Australia, November 15-19, 2021, pp. 917–929. IEEE, 2021.

[3] Prateek Munjal, Nasir Hayat, Munawar Hayat, Jamshid Sourati, and Shadab Khan. Towards robust and reproducible active learning using neural networks. CoRR, abs/2002.09564, 2020.

[4] Sudhanshu Mittal, Maxim Tatarchenko, Özgün Çiçek, and Thomas Brox. Parting with illusions about deep active learning. CoRR, abs/1912.05361, 2019.


### **CQ 2: Evaluation on additional tasks such as NLP.**

In this paper, we limit the scope of our paper to the image classification task. We have revised our paper to point this out. Evaluating active learning methods on other application domains (e.g., the NLP domain) would be very interesting. We shall consider it as future work.

---

### Decision · Program_Chairs · 2023-01-20

**Decision:**

Reject

**Justification For Why Not Higher Score:**

The paper shows an interesting empirical study of active learning, but it should include a large number of datasets in terms of classes, size, examples, etc. Also the paper should be tackle other domains like NLP. Therefore, the paper should include a more extensive evaluation before being accepted.

**Justification For Why Not Lower Score:**

N/A

**Metareview: Summary, Strengths And Weaknesses:**

# Summary
The paper conducts a review of 19 methods for supervised and semi-supervised active learning and benchmark their performance on three small CV datasets (MNIST, CIFAR, GTSRB). The paper shows that compared to supervised active learning, a semi-supervised active learning (SSAL) approach is consistently better (with RandomSampling being a baseline which many of the SAL methods underperform).  After benchmarking, it concluded that the semi-supervised active learning techniques does a much better job than SAL and traditional SAL is not always better than even random sampling.
# Strengths:
- The paper is clearly written.
- Several relevant methods have been compared and key insights clearly highlighted. It's quite interesting and meaningful. We can see from the comparative experiments that SSL does promote the AL performance.
# Weaknesses:
- The datasets are toy datasets
- Limited to Computer Vision datasets; hence it would be very interested in seeing this evaluation in the context of NLP (where data augmentation has limited benefits compared to vision) and other type of datasets.
- An important question here is regarding the re-implementation of all methods - some of the methods have made the code publically available, and such re-implementation might be unfair (both in performance and cost)